# NOMA and UAV Scheduling for Ultra-Reliable and Low-Latency Communications

**Xiaowu Liu [1], Xihan Xu [1] and Kan Yu [2,\*]**

[1] School of Information Science and Engineering, Qufu Normal University, Qufu 273165, China
[2] Key Laboratory of Universal Wireless Communications, Ministry of Education, Beijing University of Posts and Telecommunications, Beijing 100876, China
[\*] Correspondence: kanyu@bupt.edu.cn

**Abstract:** Ultra-reliable and low-latency communications (uRLLC) has received great attention in the study of wireless communication for it can provide high network performance in terms of reliability and latency. However, the reliability requirements of uRLLC require further investigation due to the inherent openness of the wireless channel. Different from the previous reliable contributions that focused on the retransmission mechanism, in this paper, we consider scenarios with the interference of multiple UAVs. We establish an analytical framework of the packet error rate (PER) for an air-to-ground (A2G) channel. In this framework, the cellular users are allocated to different UAVs according to their minimum path loss with the aim of minimizing the PER. Furthermore, a wireless link scheduling algorithm is proposed to enhance the reliability between the UAV and cellular user. Simulated results show that, under the same power and channel block length level, our proposed non-orthogonal multiple access (NOMA) scheduling scheme has the best performance.

**Keywords:** uRLLC; UAV scheduling; packet error probability; NOMA





## 1. Introduction

Ultra-reliable and low-latency communications (uRLLC) are designed to meet the requirements of reliability and delay sensitive applications, such as unmanned driving, telemedicine, satellite communication, and occupies a core position in the next generation of wireless communication. Specifically, the reliability is expressed by the packet-loss rate, where the packet-loss rate does not exceed $\leq 10^{-5}$, and the end-to-end delay is not greater than $\leq 1$ ms.

However, on the one hand, the low delay is mainly realized by the short-packet-encoding mechanism, and the conclusion based on the information theory assumption cannot be established under this mechanism—that is, when the receiver meets the decoding conditions, the decoding error probability is no longer arbitrarily small. Short packet coding affects the transmission reliability of uRLLC [1]. On the other hand, transmission path loss and channel fading are also important factors affecting uRLLC [2]. Therefore, it is urgent to investigate a new transmission mode to find a compromise between the transmission reliability and low delay under the premise of a short-packet-encoding mechanism.

With flexible mobility and hovering capability, an unmanned aerial vehicle (UAV) can provide air-to-ground line-of-sight (LoS) links, resist the loss of reliable performance caused by severe path loss and provide potential solutions for achieving ultra reliable wireless communication [3,4]. A UAV can function as a relay when there is not a strong direct communication link between two nodes. The channel quality between a UAV and ground users can be increased as a result of the high likelihood of a short LoS link, and this is seen to be crucial for meeting the latency and reliability requirements of uRLLC [5]. Therefore, we consider using UAVs as relays to provide uRLLC services for users.

In order to ensure the reliability of wireless communication and improve the spectrum utilization, Nonorthogonal Multiple Access (NOMA) has been introduced. NOMA is a

promising technology to tackle the issue of resource allocation [6]. NOMA can simultaneously serve multiple users and support large-scale connections in the same resource block. Therefore, NOMA has higher spectral efficiency compared with the Orthogonal Multiple Access (OMA) [7,8]. NOMA also allows multiple users to superimpose their signals, which may increase the transmission rate of the network.

A receiver is able to recover its required information from the superimposed signal depending on the Succession Interference Cancellation (SIC). Through SIC, NOMA can also provide reliability for wireless communication based on UAV, the fusion of NOMA and UAV has become a research hotspot in commercial and academic fields, which may significantly improve the performance of wireless networks [9–11].

Although NOMA can use SIC to obtain the required signal, achieving reliability and lower delay based on UAV and NOMA faces some challenging problems. The interference may become particularly serious particularly in large-scale traffic and high-density communications, which are typical characteristics of 5G or future 6G networks [12,13]. In order to meet these challenges, new scheduling and multiple access technologies are needed to eliminate the interference between parallel transmissions and improve the performance of wireless communication systems.

The link scheduling problem, which arises when links cooperate well in the presence of noise and interference, is a major determinant of wireless network communication effectiveness. An effective link scheduling scheme can greatly reduce the interference caused by concurrent links. It is worth noting that the selection of interference model determines the complexity of the algorithm design and complexity in wireless link scheduling. In general, contemporary contributions heavily rely on three interference models: the graph-based model, the signal-to-interference-noise ratio (SINR) model and the Rayleigh fading model [14–17].

Based on the above observations, consider a scenario where the UAV acts as a relay to forward base station (BS) information to the cellular user (CU). As CU is interfered with by other UAVs, we design a UAV scheduling technology to avoid CU being interfered with by non-associated UAVs. Then, the error probability is minimized on the basis of NOMA. The main contributions of this paper are as follows.

- By assuming that UAVs operate in crowded cities, we describe the statistical properties of SINR of CU under a three-dimensional model. The average path loss of the LoS and NLoS links is computed in accordance with the potential of constructing a LoS link between the UAV and CU, and the SINR expression of the CU may be obtained.
- In order to avoid the interference of non-associated UAVs on CU, we introduce link scheduling technology. Through the distributed DLS algorithm, we propose an algorithm for scheduling UAV according to DLS, which determines the switching state of each UAV and minimizes the interference level between parallel transmissions.
- In this paper, we introduce NOMA into uRLLC. To reduce the likelihood of an error, we maximize the power management and allocation of the total channel block length. Given the limitations of service quality, we optimized the resource allocation scheme from UAV to CU.

The paper is structured as follows. Section 2 reviews the related work. Section 3 describes the network model. Section 4 introduces the UAV-scheduling scheme and the resource allocation scheme. Section 5 presents our analysis simulation experiments. Section 6 concludes the paper.

## 2. Related Work

The study on uRLLC mainly focuses on a terrestrial wireless network, and the UAV-assisted uRLLC is still a novel mechanism that emerged in recent years. For the implementation of uRLLC with UAV, a framework was established to realize the ultra reliable transmission and to meet the requirement of low delay [18].

Its efficacy in control and non-payload communication (CNPC) links was confirmed. As the crucial metrics for the UAV communication system, the average packet error rate

(PER) of control links was studied, and the closed expressions of these indexes were also derived in [19]. In particular, the average data rate from the ground station to UAV is also discussed under a three-dimensional channel model.

NOMA is important for the implementation of uRLLC because it reduces transmission error probability and transmission delay. However, the dependability of NOMA limits its application in uRLLC. In terms of implementing uRLLC based on NOMA, the performance of the serving uplink uRLLC system under NOMA was studied, and the average power consumption of each packet was minimized under the constraint of uRLLC [20]. In [21], the best resource distribution plan for uRLLC's uplink (UL) and downlink (DL) NOMA was investigated. Zhai et al. studied the joint UAV power, channel and height-optimization strategy in [22].

Many centralized algorithms for maximum link scheduling have been proposed [23–27], and these algorithms can improve the link availability in a wireless communication system. In recent years, an approximation technique for MLS was developed, and Goussevskaia et al. gave NP-completeness proofs for maximum link scheduling (DLS) in the SINR model [24]. Goussevskaia et al. introduced an algorithm for MLS in [28] that is independent of the network topology and has a constant approximation ratio. These contributions provide a novel mechanism to study the scheduling issue in a UAV-assisted wireless network.

In [14], Dams et al. suggested link scheduling methods that were appropriate for both the Rayleigh fading model and the SINR model. The authors showed that many scheduling algorithm in the SINR model may be transformed into the Rayleigh fading model. This can improve the analysis ability of the Rayleigh fading model without losing the practicality of the Rayleigh fading model. In [29–32], the graph-based model and SINR model have both been extensively used to study link scheduling.

In contrast, research on link scheduling with the Rayleigh fading model is much less extensive. Additionally, the Rayleigh fading model's condition for link scheduling success in a SINR-feasible set is at least $1/e$. The number of successful transmissions converges to a fixed portion of the non-fading optimum in the case of average power allocation. They demonstrated this by using a proper learning algorithm to solve the MLS problem. These MLS problem algorithms are able to determine the number of scheduling links since they use polynomial rounds to converge in distributed situations [33].

However, there are few studies on link scheduling and NOMA to implement uRLLC. To work on improving the availability of the network, this paper concentrates on the situation that different UAVs serve multiple users in the same network. Link scheduling technology is used to lessen the interference that UAVs cause to one another simultaneously.

## 3. Network Model

As shown in Figure 1, we consider a downlink urban cellular network with UAV support that consists of a BS, $u$ UAVs and $n$ CUs. In order to establish a LOS link between the BS and CU, the BS sends the signal to the UAV, and the UAV forwards the signal to the CU, where the UAV hovers above the CU as a relay. As shown in Figure 1, $UAV_1$ sends signals to $CU_1$ and $CU_2$, and $UAV_2$ sends signals to $CU_3$ and $CU_4$. When the $CU_2$ is close to $UAV_2$, it will be interfered with by the $UAV_2$. Therefore, we eliminate the interference caused by non-associated UAVs to the CU through a UAV-scheduling algorithm, then eliminate the interference caused by the same UAV through SIC and, finally, realize the Quality of Service (QoS) requirements of uRLLC by optimizing the power and channel block length.

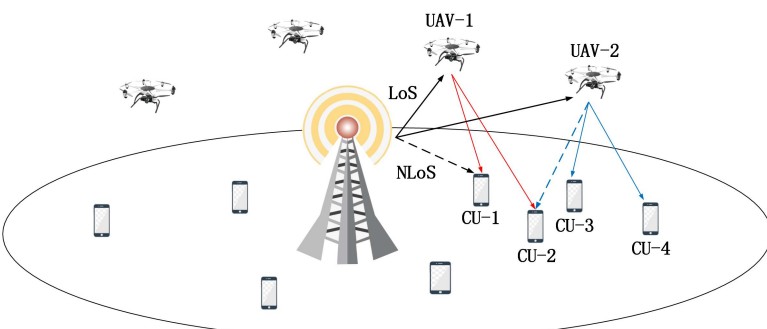

**Figure 1.** Network model.

### 3.1. Channel Model

In the channel model, the distance from the BS to the UAV is much greater than that from the UAV to the CU. Therefore, we mainly analyze the communication channel from UAV to CU. In this model, considering the urban scene with obstacles, the likelihood of a LoS communication link being established between the UAV and CU is $P_{\text{LoS}}$. The likelihood of establishing a LoS communication link is given by [34]

$$P_{\text{LoS}} = \frac{1}{1 + a\exp(-b(\theta - a))} \tag{1}$$

where $\theta$ is the elevation angle between the UAV and the CU, and $a$ and $b$ are constants based on the type of communication environment. As a result, we can conclude that the probability of NLoS is $P_{\text{NLoS}} = 1 - P_{\text{LoS}}$.

The following is an expression for the channel path loss model of LoS and NLoS links

$$L_k = 20\lg\left(\frac{4\pi f_c d}{c}\right) + \eta_k, \text{k} \in \{\text{LoS}, \text{NLoS}\} \tag{2}$$

where path loss in free space is the first term and extra path losses of LoS and NLoS are $\eta_{\text{LoS}}$ and $\eta_{\text{NLoS}}$, respectively. We consider the average path loss for a fixed UAV based on determining the likelihood of LoS link

$$L(\theta, d) = L_{\text{LoS}}P_{\text{LoS}} + L_{\text{NLoS}}P_{\text{NLoS}} \tag{3}$$

The average path loss in Equation (3) can be rewritten by inserting Equations (1) and (2) into Equation (3).

$$L(\theta, d) = \frac{A}{1 + a\exp(-b(\theta - a))} + 20\lg(d) + C \tag{4}$$

where $A = \eta_{\text{LoS}} - \eta_{\text{NLoS}}, C = 20\lg\left(\frac{4\pi f_c}{c}\right) + \eta_{\text{NLoS}}$

The transmission power of UAV is $P^u$, and the noise power at CU is $\sigma^2$. $I$ indicates the interference to ground CU. The $\gamma$ of CU is expressed as

$$\gamma = \frac{P^u g}{I + \sigma^2} \tag{5}$$

where $\gamma$ represents the SINR of CU, and $g = (1/L(\theta, d))$ represents the channel gain between UAV and CU.

### 3.2. QoS Requirements

In uRLLC, QoS has strict requirements on $D_{max}$ and $\varepsilon_{max}$, where $D_{max}$ represents the maximum transmission delay, and $\varepsilon_{max}$ denotes the maximum decoding error probability

allowed by CUs. The percentage of data bits used by each channel to the total bits is how we define the coding rate $R$. In general, the Shannon capacity can accomplish exceptionally low packet-error rates for a sufficient number of codewords. However, in uRLLC, in order to ensure strict delay requirements using short packets for transmission, the Shannon capacity formula cannot be used.

We suppose that the size of packet is $L$ bits and that the transmission is completed in $T$ seconds. The total number of bits utilized by the channel can be written as $M = B \cdot T$, where $B$ is the system's bandwidth. Therefore, the coding rate $R = L/M$. To meet the QoS requirements of uRLLC, short packet transmission is required; therefore, the transmission rate can be approximately [35]

$$R \approx \log_2(1 + \gamma) - \sqrt{\frac{V}{M}} \frac{Q^{-1}(\varepsilon)}{\ln 2} \tag{6}$$

where $\varepsilon$ is the decoding error probability at the CU, $V = 1 - (1 + \gamma)^{-2}$ is the channel dispersion, and $Q^{-1}(\varepsilon)$ is the reciprocal of the Gaussian Q function. Received $\gamma$ is usually higher than 5 dB in uRLLC, and so $V(\gamma) \approx 1$. For a fixed $R$, the decoding error probability in uRLLC transmission can be approximately

$$\varepsilon = Q(f(\gamma, M, L)) \tag{7}$$

where $f(\gamma, M, L) = \sqrt{M}(\ln(1 + \gamma) - R_s), R_s = \ln 2 \cdot (L/M)$. Consequently, in order to fulfill the reliability criteria, it is essential to

$$\varepsilon \leq \varepsilon_{max} \tag{8}$$

*3.3. Problem Formulation and Solution*

Due to the introduction of multiple UAVs in the network, the interference suffered by CUs will also increase. We employ link scheduling technologies to schedule UAVs in order to lessen the interference that CUs experience and to achieve uRLLC. Suppose that $E$ is a concurrent scheduling link set in the same time slot. It is anticipated that the receiver $CU_i$ will be able to effectively decode the signal of the sender $UAV_i$ if the SINR is larger than or equal to the threshold $\beta$. SINR received by $CU_i$ is as follows:

$$Y_{l_i, E} = \frac{g_{ii} \cdot d_{ii}^{-\lambda}}{\Sigma g_{ji} \cdot d_{ji}^{-\lambda} + \omega} \geq \beta \tag{9}$$

where $l_i$ indicates the link from $UAV_i$ to $CU_i$, $g_{ji}$ is the channel fading gain and $d_{ji}$ is the distance between $UAV_j$ and $CU_i$, $\lambda$ represents the path loss index, and $v = \omega P$ means the noise and intra group interference.

From Equation (7), we can see that the decoding error rate is not only related to SINR but also related to $M$. The transmission must be finished within $M$ symbols or channel usage if the E2E delay is calculated using the length of the channel block, where $M = D_{max} \cdot B$. E2E delay requirements can be written as $m^u + m^c + m^q \leq M$, where $m^u$, $m^c$ and $m^q$ are the channel block lengths needed by the BS to communicate with the UAV, UAV to CU and queuing, respectively. In addition, the requirement for the overall error probability can be expressed as $1 - (1 - \varepsilon^u)(1 - \varepsilon^c)(1 - \varepsilon^q) \approx \varepsilon^u + \varepsilon^c + \varepsilon^q \leq \varepsilon_{max}$, where

$\varepsilon^u$, $\varepsilon^c$ and $\varepsilon^q$ is the error probability of the BS to UAV, UAV to CU and queuing, respectively. The optimization problem is formulated as

$$\min_{Y,M,\varepsilon_{max}} \varepsilon^* \tag{10}$$

$$\text{subject to} : Y \geq \beta \tag{10a}$$

$$D^* \leq D_{max} \tag{10b}$$

$$m^u + m^c + m^q \leq M \qquad\qquad m^u, m^c, m^q \in \mathbb{Z} \tag{10c}$$

$$\varepsilon^u + \varepsilon^c + \varepsilon^q \leq \varepsilon_{max} \tag{10d}$$

where $\varepsilon^*$ is the overall error probability, $D^*$ is the transmission delay, and (10a) is the precondition for successful decoding of CU. According to the above, (10c) can be used as the delay constraint of transmission, and (10d) is the constraint on the overall decoding error rate.

## 4. UAV Scheduling and Resource Allocation Scheme

In this section, we first associate the UAV and CU according to the path loss to achieve reliability. Then, we propose a UAV-scheduling scheme based on the SINR model to reduce interference and enable the CU to decode the received signal.

### 4.1. UAV-Scheduling Scheme

In order to obtain good channel coefficients between UAV and CU, we propose the Algorithm 1 to associate UAV and CU according to the minimum path loss. First, each $CU_j$ selects the $UAV_{i_j}$ with the smallest path loss for communication. $PL_{ji}$ indicates the path loss from $UAV_i$ to $CU_j$. After all CUs select UAVs, each UAV will generate an associated CU list. The CU list of $UAV_i$ is represented by $\zeta_i$. If the CU list of UAV is empty, it will be removed from the network.

---

**Algorithm 1** UAV-CU association scheme.

---

1: Given $u$ UAVs and $n$ CUs
2: **for** $j = 1$ to $n$ **do**
3:    **for** $i = 1$ to $u$ **do**
4:       **if** $PL_{ji}$ is the smallest **then**
5:          add $CU_j$ into $\zeta_i$.
6:       **end if**
7:    **end for**
8: **end for**
9: **if** $\zeta_i$ without CU **then**
10:    Remove $UAV_i$ from network.
11: **end if**
12: Output:$\zeta_i$

---

Suppose that, in the SINR model, when the probability of Equation (9) failure is less than $\epsilon$, the link between $UAV_i$ and $CU_i$ is successfully scheduled, where $\epsilon$ indicates the acceptable probability of transmission failure. In other words, if the probability of success of all links in set E is greater than $1 - \epsilon$, then we consider set E to be SINR-$1 - \epsilon$-feasible. In order to successfully eliminate the interference and improve the number of successful links, we propose a UAV-scheduling algorithm.

The relative interference (RI) of link $l_j$ on link $l_i$ is caused by $l_j$ in the inverse of the SINR at $l_i$—namely, $RI_j(i) = \frac{1/d_{ji}^\lambda}{1/d_{ii}^\lambda}$. The impact of link $l_i$ can be seen as being caused by the links in a set S with power 1. Combined with the sum of link related interference in s, the impact on $l_j$ can be expressed as $c_i$.

$$a_S(l_j, l_i) = \sum a(l_j, l_i) = \sum c_i \cdot RI_j(i) \tag{11}$$

where $c_i = \frac{1}{1-\beta\omega d_{ii}^\lambda}$.

Generally speaking, in order to improve the success probability, there are usually two methods. First, by preventing the transmission of some successful links surrounding a receiver, the success probability for each link in the SINR feasible set can be raised to $1 - \epsilon$. Another method is to consider only a certain range of interference. Based on the above two points, only when the impact of the chosen connection is below threshold $c$, which is specified by the parameters of the path loss index $\lambda$ and SINR threshold $\beta$, can we be certain that link $l_i$ will be scheduled. For the successfully scheduled link $l_i$, the impact of the previously successfully scheduled link to $l_i$ must meet the following conditions.

$$c_i \sum_{l_i \in E} RI_j(i) = c_i \sum_{l_i \in E} \frac{1/d_{ji}^\lambda}{1/d_{ii}^\lambda} \leq c \tag{12}$$

Thus, we can obtain

$$d_{min}^1 > \left( \frac{d_{ii}^\lambda}{1-\beta\omega d_{ii}^\lambda} \cdot \frac{1}{(\lambda - 1)c} \right)^{\frac{1}{\lambda-1}} \tag{13}$$

In addition, if the link $l_j$ is successfully scheduled and not the last, it must meet

$$c_j \sum_{l_j \in E, d_{ii} < d_{jj}} \frac{1/d_{ij}^\lambda}{1/d_{jj}^\lambda} + c_j \sum_{l_k \in E, d_{kk} < d_{jj}} \frac{1/d_{kj}^\lambda}{1/d_{jj}^\lambda} \leq \frac{1}{\beta} \tag{14}$$

We find

$$d_{min}^2 > \left( \frac{d_{ii}^\lambda}{1-\beta\omega d_{ii}^\lambda} \cdot \frac{1}{(\lambda-1)(1/\beta - c)} \right)^{\frac{1}{\lambda-1}} \tag{15}$$

The distance between the scheduling links before and after the link $l_i$ is at least $d_{min}$, providing that the connection $l_i$ is successfully scheduled.

$$d_{min} = \left( \frac{d_{ii}^\lambda}{1-\beta\omega d_{ii}^\lambda} \cdot \frac{\lambda}{(\lambda-1)c} \right)^{\frac{1}{\lambda-1}} \tag{16}$$

In light of the distance constraint given in Equation (16), we further conclude that the distance between the non-associated $UAV_j$ and the $CU_i$ is at least $\delta\varphi d_{min}$, where $\delta = (\beta c)^{\frac{1}{\lambda-1}}$ and $\varphi = (\frac{1}{\ln\frac{1}{1-\epsilon}})^{\frac{1}{\lambda-1}}$.

To satisfy the SINR constraint, we set the switch activation mode of UAV—that is $\{\theta_i(t)\}_{i=1}^N$, where $\theta_i(t) \in \{0, 1\}$ indicating whether the $l_i$ is active within the scheduling interval $t$. If $UAV_i$ and $CU_{i_j}$ can communicate within scheduling interval $t$, then $\theta_i = 1$ and, otherwise, $\theta_i = 0$. Setting an acceptable send probability q, which ensures that the link can successfully exchange messages with a given probability, can be used to manage the rivalry among UAVs. The scheduling message $m_s$ is sent to those unanticipated CUs of $\delta\varphi d_{min}$ when the $UAV_i$ is scheduled to communicate with its corresponding $CU_{i_j}$.

Then, the CUs sends messages to its corresponding UAVs in probability $p_t$, and the UAVs that receives the message exits the current scheduling process. Thus, for each selected link in $E$, the distance between CU and all unexpected UAVs is at least $\delta\varphi d_{min}$ by using two broadcasts.

Through Algorithm 2, we find that the impact on $CU_i$ is

$$\sum a(l_j, l_i) = c_i d_{ii}^\lambda \sum_{l_j \in E} \frac{1}{d_{ji}^\lambda}$$

$$< c_i d_{ii}^\lambda \frac{\lambda}{(\lambda - 1)c} \left( \frac{1}{\delta \varphi d_{min}} \right)^{\lambda - 1}$$

$$= \frac{d_{ii}^\lambda}{1 - \beta \omega d_{ii}^\lambda} \cdot \frac{\lambda}{\lambda - 1} \cdot \frac{1}{\frac{d_{ii}^\lambda}{1 - \beta \omega d_{ii}^\lambda} \cdot \frac{1}{\ln\left(\frac{1}{1-\epsilon}\right)} \frac{\lambda \beta}{(\lambda - 1)}} \qquad (17)$$

$$= \frac{1}{\left(\frac{1}{\ln \frac{1}{1-\epsilon}}\right)\beta}$$

It has been proven that, if $l_i$ received SINR in accordance with the SINR model, it would be at least $\left(\frac{1}{\ln \frac{1}{1-\epsilon}}\right)\beta$, $l_i$ can be successfully scheduled [15]. The influence received by $CU_i$ is the inverse of SINR at $CU_i$. Therefore, when the distance between the $CU_i$ and other sender $UAV_j$ is at least $\delta \varphi d_{min}$, the link $l_i$ can fulfill the given SINR requirements and can be successfully scheduled.

---

**Algorithm 2** UAV-scheduling scheme.

---

1: Given $n$ links, $u$ UAVs and $n$ CUs
2: Assign transmission probability $p_t = \frac{1}{n}$ for each UAV
3: **if** A UAV starts to transmit with $p_t = \frac{1}{n}$ **then**
4:     It broadcasts $m_s$ to CUs within $\delta \varphi d_{min}$.
5:     **if** A CU only receives a $m_s$ from its associated UAV **then**
6:         It broadcasts $m_{s2}$ to UAVs within $\delta \varphi d_{min}$
7:     **end if**
8: **end if**
9: **if** $UAV_i$ receives $m_{s2}$ from non-associated CUs **then**
10:     $\theta_i = 0$
11: **end if**
12: **if** $UAV_i$ receives $m_{s2}$ from its intended CU **then**
13:     $\theta_i = 1$
14: **end if**
15: output:$\theta_i$

---

*4.2. Resource Allocation Scheme*

4.2.1. Transmission Scheme from UAV to CU

In NOMA, the UAV sends signals to CUs with different power in the same resource block. Therefore, we can obtain $m_{cu_1} = m_{cu_2} = m^c$, where $m_{cu_1}$ and $m_{cu_2}$ are the channel block lengths allocated to $CU_1$ and $CU_2$, respectively. The transmission signal at the UAV is $\Sigma_{i=1}^{2} \sqrt{\alpha_i P^u} x_i$, where $x_i, i = 1, 2$ is the signal to $CU_i$, $P^u$ is the overall transmission power of the UAV, and $\alpha_1$ and $\alpha_2$ are the power allocated to different CUs, respectively, which satisfy $\alpha_1 + \alpha_2 = 1, \alpha_1 \leq \alpha_2$.

$$y_i = \sqrt{g_i} s + n_i = \sqrt{g_i}\left(\sqrt{\alpha_i P^u} x_1 + \sqrt{\alpha_i P^u} x_2\right) + n_i \qquad (18)$$

$n_i^d \sim \mathcal{CN}(0, \sigma_i^2)$ is the complex additive white Gaussian noise. For ease of use, we set $\sigma_1^2 = \sigma_2^2 = \sigma^2$.

At $CU_1$, $x_2$ is decoded first, and then we decode $x_1$ with successive interference cancellation (SIC). The SINR of decoding $x_2$ is given by the following formula

$$\gamma_2^1 = \frac{\alpha_2 P^u g_1}{\alpha_1 P^u g_1 + \sigma^2} = \frac{\alpha_2 g_1}{\alpha_1 g_1 + 1/\rho} \qquad (19)$$

where $\rho = P^u / (\sigma^2)$. The error rate of $x_2$ decoding at CU$_1$ is $\varepsilon_2^1 = Q(f(\gamma_2^1, m^c, L))$.

If CU$_1$ successfully decodes $x_2$, CU$_1$ can decode $x_1$ without intra-group interference. The SINR of decoded $x_1$ is $\gamma_1^1 = \alpha_1 g_1 \rho$. The probability of decoding error for $x_1$ at CU$_1$ is given by $\varepsilon_1^1 = Q(f(\gamma_1^1, m^c, L))$.

If SIC fails, CU$_1$ can still attempt to use SINR $\widehat{\gamma}_1^1 = \alpha_1 g_1 / (\alpha_2 g_1 + 1/\rho)$ decode $x_1$. We find that $\widehat{\gamma}_1^1$ is too small to fulfill the QoS of uRLLC. Therefore, in this example, we suppose that the error probability of directly decoding $x_1$ at CU$_1$ is 1. Furthermore, the average decoding error probability at CU$_1$ can be approximately

$$\varepsilon^1 = (1 - \varepsilon_2^1)\varepsilon_1^1 + \varepsilon_2^1 \approx \varepsilon_1^1 + \varepsilon_2^1 \tag{20}$$

CU$_2$ directly decodes $x_2$, and the received SINR is

$$\gamma_2^2 = \frac{\alpha_2 g_2}{\alpha_1 g_2 + 1/\rho} \tag{21}$$

The corresponding decoding error probability at CU$_2$ can be given by the following formula: $\varepsilon^2 = \varepsilon_2^2 = Q(f(\gamma_2^2, m^c, L))$.

### 4.2.2. Transmission Scheme from BS to UAV

In the transmission from BS to UAV, there is an air-to-air LoS link between BS and UAV. We assume that the UAV will not receive other interference while receiving the signal.

The received SNR at the UAV is

$$\gamma^u = \frac{P^b |\widetilde{h}^b|^2}{\sigma_u^2} = |\widetilde{h}^b|^2 \rho^u \tag{22}$$

where $\rho^u$ represents the power sent by BS to UAV, $\rho^u = P^b / \sigma_u^2$, $\widetilde{h}^b$ represents the channel coefficient from BS to UAV. Therefore, the probability of decoding error at the UAV is determined by $\varepsilon^u = Q(f(\gamma^u, m^u, L))$ given.

### 4.2.3. Queuing Scheme

In uRLLC, the queuing delay $D^q$ cannot be greater than 1 ms, and, when the delay is fixed, the service rate is also fixed. Therefore, the queuing delay requirement can be expressed in terms of the effective bandwidth. For example, the fixed service rate of UAV should be greater than or equal to $E^B$. In NOMA, we use a queue to make the packets sent by UAV to CUs wait in the same buffer.

The effective bandwidth according to Poisson arrival process is as follows:

$$E^B = \frac{L \ln(1/\varepsilon^q)}{m^q \ln\left(1 + \frac{\ln(1/\varepsilon^q)}{\delta m^q}\right)} \tag{23}$$

where $\delta$ is the average packet arrival rate, and $m^q$ is the channel block length required for queuing. Therefore, if $L/m^q > E^B$, it is possible to ensure the queuing delay requirement $(D^q, \varepsilon^q)$.

The overall error probability from BS to CU$_1$ is given by the following formula

$$\varepsilon_{\text{BS,CU}_1} = \varepsilon^u + \varepsilon_1^1 + \varepsilon_2^1 + \varepsilon^q \tag{24}$$

Furthermore, from BS to CU$_2$ is given by the following formula

$$\varepsilon_{\text{BS,CU}_2} = \varepsilon^u + \varepsilon_2^2 + \varepsilon^q \tag{25}$$

We aim to minimize $\varepsilon_{\text{BS,CU}_1}$ when $\varepsilon_{\text{BS,CU}_2}$ is not greater than the threshold $\varepsilon^{th}$. Therefore, the problem of resource allocation can be expressed as

$$\min_{m^u, m^q, m^c} \varepsilon_{\text{BS,CU}_1} \tag{26}$$

$$\varepsilon_{\text{BS,CU}_2} \le \varepsilon^{th} \tag{26a}$$

$$E^B \le L/m^q \tag{26b}$$

$$m^u + m^q + m^c \le M \tag{26c}$$

$$0 < \alpha_i < 1, \alpha_1 + \alpha_2 \le 1 \tag{26d}$$

where (26a) is the constraint of the probability of total error from BS to $CU_2$, (26b) is the constraint of the queuing delay requirement, (26c) is the restriction of the overall channel block length, and (26d) is the constraint of the total transmission power.

**Lemma 1.** *Constraint (26c) is always equal at the optimal solution.*

**Proof.** We demonstrate that $\varepsilon^q$ strictly reduces $m^q$ by using the formula $g(m^q) = m^q \ln(1 + \ln(1/\varepsilon^q)/(\delta m^q))$ through the first order and second order derivatives of $g(m^q)$. Then, we can prove that constraint (26c) is always equal at the optimal solution because $\varepsilon^u, \varepsilon_1^1, \varepsilon_2^1$ decreases linearly with the corresponding channel blocklength. □

**Lemma 2.** *Constraint (26b) is always equal at the optimal solution.*

**Proof.** We find $E^B$ based on $(m^q, \varepsilon^q)$ in (23). Therefore, we can determine $m^q$ first, and then we simply need to analyze $E^B$ by $\varepsilon^q$ $E^B$. The proof of Lemma 1 states that $E^B$ is a monotone decreasing function of $\varepsilon^q$. Therefore, as $E^B$ increases, $\varepsilon^q$ decreases as well as the value of $\varepsilon_{\text{BS,CU}_1}$. □

Lemma 2 demonstrates that $\varepsilon^q$ can be uniquely determined when $m^C$ and $m^q$ are determined. Then, we want to reduce the values of $\varepsilon^1$ and $\varepsilon^u$. We have $\varepsilon_2^2 + \varepsilon^u \le \varepsilon^{th} - \varepsilon^q \equiv \varepsilon^{th(1)}$ as a result of rearranging constraint (26a) to become $\varepsilon_2^2 \le \varepsilon^{th(1)}$ and $\varepsilon^u \le \varepsilon^{th(1)}$. First, we analyze $\varepsilon^1$ without the restriction $\varepsilon_2^2 \le \varepsilon^{th(1)}$.

By $Q(0) = 0.5$, we can find $\log_2(1 + \gamma_i^1) \ge L/m^c, i = \{1, 2\}$. Consequently, it is possible to determine the minimum values of $\alpha_1$.

$$\alpha_1^{lb} = \frac{2^{L/m^c} - 1}{g_1 \rho} \tag{27}$$

$$\alpha_1^{ub} = \frac{g_2 \rho - 2^{L/m^c} + 1}{2^{L/m^c} g_1 \rho} \tag{28}$$

**Lemma 3.** *$\varepsilon^1$ first strictly decrease and then increase with respect to $\alpha_1$ in $(\alpha_1^{lb}, \alpha_1^{ub})$.*

**Proof.** It is simple to confirm that there is a distinct $\alpha_1'$ by the first order derivative of $\varepsilon^1$ with regard to $\alpha_1$, which makes $\varepsilon^1$ strictly decrease on $(\alpha_1^{lb}, \alpha_1')$ and strictly increase on $(\alpha_1', \alpha_1^{ub})$. □

Lemma 3 demonstrates that the least valuable $\alpha_1'$ of $\varepsilon_1^1$ may be located using the one-dimensional linear search approach based on a dichotomy search.

Then, we analyze $\varepsilon^u$. When $P^b$ and $L$ are fixed, $\varepsilon^u$ is only related to $m^u$.

Finally, finding the lower bound $\varepsilon^1$ under the restriction of $\varepsilon^2 \le \varepsilon^{th(1)}$ is our objective. As $\gamma_2^2$ decreases strictly with $\alpha_1$, $\varepsilon_2^2$ increases with the increase of $\alpha_1$. Therefore, the maximum value of $\alpha_1$ can be obtained by $\varepsilon^2 = \varepsilon^{th(1)}$. We suppose $\alpha_1^{opt}$ is the ideal method for

allocating power. If $\alpha_1^{th} \geq \alpha_1'$, we have $\alpha_1^{opt} = \alpha_1'$. If $\alpha_1^{th} \leq \alpha_1'$, we have $\alpha_1^{opt} = \alpha_1^{th}$—that is, $\alpha_1^{opt} = \min\{\alpha_1', \alpha_1^{th}\}$.

Next, our objective is to identify the ideal channel-block-length-allocation plan.

We first consider the minimum value of $m^u$, $m^c$ and $m^q$. From (26a), we have $\varepsilon^u \leq \varepsilon^{th}$, $\varepsilon^2 \leq \varepsilon^{th}$, $\varepsilon^q \leq \varepsilon^{th}$. Therefore, we have

$$\varepsilon^{th} \geq \varepsilon^u = Q\left(\ln 2\sqrt{m^u}(\log_2(1 + |\tilde{h}^b|^2 \rho^u) - L/m^u)\right) \tag{29}$$

by settling $\varepsilon^{th} = \varepsilon^u$, the minimum value $m^{u,lb}$ of $m^u$ can be solved.

$$\begin{aligned}
\varepsilon^{th} &\geq \varepsilon^2 \\
&= Q\left(\ln 2\sqrt{m^c}\left(\log_2\left(1 + \frac{\alpha_2 g_2}{\alpha_1 g_2 + 1/\rho}\right) - L/m^c\right)\right) \\
&\geq \varepsilon^{2,lb} = Q\left(\ln 2\sqrt{m^c}(\log_2(1 + g_2\rho) - L/m^c)\right)
\end{aligned} \tag{30}$$

through solving $\varepsilon^{th} = \varepsilon^{2,lb}$, the lower bound $m^{c,lb}$ of $m^c$ can be solved. We know that $\varepsilon^q$ strictly decreases with the increase of $m^q$ from the proof of Lemma 1. Therefore, by determining $\varepsilon^q = \varepsilon^{th}$, one may also obtain the lower bound of the value of $m^q$, denoted as $m^{q,lb}$. In this manner, the shortest overall channel block length is provided by $m^{lb} = m^{u,lb} + m^{c,lb} + m^{q,lb}$.

Then, we can use dynamic programming to solve the channel-block-length-allocation problem. Assumed state $S_m(m^{lb} \leq m \leq M)$ means that m channel block lengths are already assigned. $\varepsilon_{BS,CU_1}(S_m)$ represents the function value of state $S_m$. Therefore, we can find

$$\varepsilon_{BS,CU_1}(S_m) = \min\{\varepsilon_{BS,CU_1}(S_{m-1,m})\} \tag{31}$$

where $S_{m-1,m}$ denotes the distribution of the m-th channel block length following the allocation of the $m-1$ channel block length. Its three components, $S_{m-1,m}^u$, $S_{m-1,m}^c$ and $S_{m-1,m}^q$, show that the m-th channel block length is set aside for the transmission from BS to UAV and for the transmission from UAV to CU and queuing, respectively.

From (31), we discover that the state progress succession is a Markov chain, and the previously mentioned power control technique can resolve the $\varepsilon_{BS,CU_1}(S_{m-1,m})$. Therefore, we find the optimal channel block length scheme through the given $S_{m^{lb}}$.

### 4.3. Algorithm Analysis

After analysis, the time complexity of Algorithm 1 is $\mathcal{O}(nu)$. For the time complexity of Algorithm 2, we consider the worst case as $\mathcal{O}(n \ln n)$, and the complexity of the proposed power control and channel-block-length-allocation scheme is $\mathcal{O}\left(6M \log_2\left(\frac{1}{2\varepsilon}\right)\right)$, and $\varepsilon$ is the approximation error of the binary search. Therefore, the total complexity of our proposed algorithm is $\mathcal{O}\left(nu + n \ln n + 6M \log_2\left(\frac{1}{2\varepsilon}\right)\right)$.

## 5. Simulation and Numerical Results

In this section, we first provide the simulation results of two different scheduling schemes, including the DLS scheduling scheme and GHW scheduling scheme. The simulation displays the DLS scheduling plan's effectiveness. After determining the advantages of our proposed scheme, we propose three simultaneous interpreting schemes, including NOMA, OMA and NOMA scheduling. NOMA scheduling is a combination of NOMA and DLS.

### 5.1. MLS Simulation

We consider a random network in which $U$ UAVs and $n$ CUs are randomly distributed in a circular area with a radius of 500 m. The SINR parameter is set to $U = 5$, $\epsilon \in \{0.05, 0.1\}$, $\beta = 1.2$, $v = -100$ dB, $P = 40$ mW and $\gamma = 5$. First, as shown in Figure 2, we examine the

effect that an increase in the number of links in the random topology has on the quantity of correctly scheduled links in DLS and GHW. The graph shows that, as the number of links rises, the DLS method schedules more links than the GHW method. Due to the restriction of parameter c, the GHW cannot schedule more links. After the DLS algorithm, the CU in the link removes more interference from non-associated UAVs. Alternatively, as demonstrated in Algorithm 2, the distance between the CU and all unexpected UAVs is at least $\delta\varphi d_{min}$ and $\varphi = \left(\frac{1}{\ln\frac{1}{1-\epsilon}}\right)^{\frac{1}{\lambda-1}} . P_{opu}$.

In Figure 3, we investigate how the path loss index affects the effectiveness of the aforementioned algorithm scheduling. The simulation was run with the parameters $n = 500$ and $\epsilon = 0.1$, with all other values being the same as before. The algorithm performance increased with $\lambda$. Specifically, the DLS algorithm showed better scheduling performance than did the GHW algorithm. This is because, when the link length is fixed, with larger $\lambda$ and smaller $\delta\varphi d_{min}$, more links can be successfully transmitted according to the above algorithm. For smaller $\lambda$, the distance constraint $\delta\varphi d_{min}$ becomes larger, and more UAVs will stop sending once they receive the message $m_a$. Therefore, a small number of links can be successfully scheduled.

This is because the influence of $\lambda$ on the interference signal is better than that on the expected signal, and thus CU can obtain a greater SINR. Following the increase of $\lambda$, more links can be successfully scheduled.

Next, we use the above simulation conditions, as shown in Figure 4, to more thoroughly research how the SINR threshold affects scheduling effectiveness. With an increase in SINR, fewer successful linkages are formed. This is due to the fact that the $\delta\varphi d_{min}$ increases with the SINR. In other words, it can only be said to be successful if the CU is outside of the wider interference range of any unanticipated UAVs. Between the two algorithms, the DLS algorithm performed better than the GHW algorithm.

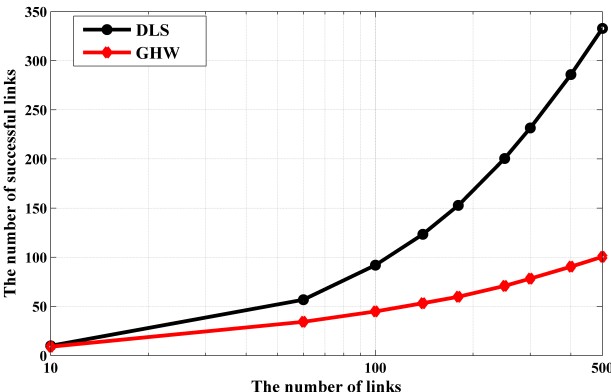

**Figure 2.** The scheduling performance effects of the number of links.

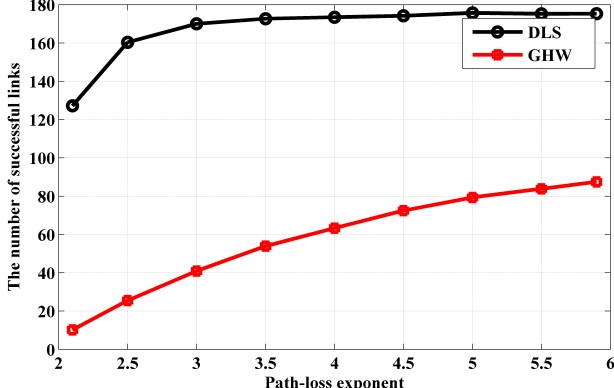

**Figure 3.** The effects of path-loss exponent on scheduling performance.

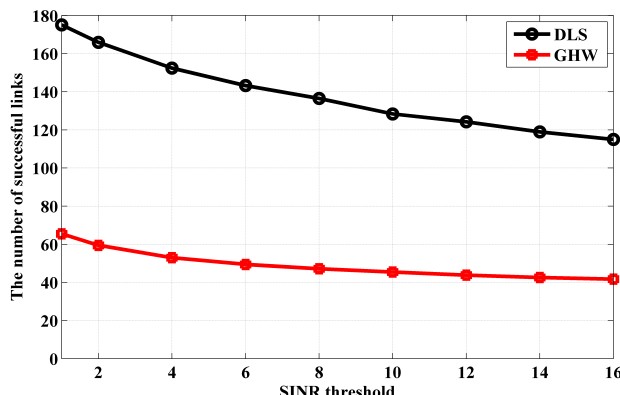

**Figure 4.** SINR threshold and link-size effects on scheduling performance.

*5.2. Packet Error Rate Simulation*

To show the effectiveness of uRLLC, we give simulation results for three alternative transmission systems, including the NOMA scheme, OMA scheme and NOMA scheduling scheme.

The simulation parameters are set as follows. Assuming $d^b = 200$ m and $d_1^u = d_2^u = 150$ m, $d^b, d_1^u, d_2^u$ is the distance between BS and UAV, UAV and $CU_1$, and UAV and $CU_2$, respectively. The system bandwidth is set to $B = 1$ MHz, so the E2E delay $D_{max} \leq 1$ ms. We set packet arrival rate and size to $\omega = 10$ packets/s and $L = 160$ bits, respectively. The noise power spectral density was set to $-173$ dbm/Hz, and the large-scale path loss was set to $35.3 + 37.6lg(d)$ dB, where $d$ is the distance between UAV and CU. The overall error probability of the target was set to $10^{-5}$, and $\varepsilon_{BS,CU_1}$ is the performance index.

In Figure 5, we study the effect of the total transmission power on $\varepsilon_{BS,CU_1}$. It can be shown that $\varepsilon_{BS,CU_1}$ decreases from 1 to $10^{-12}$ as the total transmit power grows from 0.4 to 2 W. The performance of the NOMA scheduling system is superior to the other two schemes. This is because the CU in the NOMA scheduling scheme not only eliminates the interference from other UAVs but also shares the channel block length and reduces the error rate.

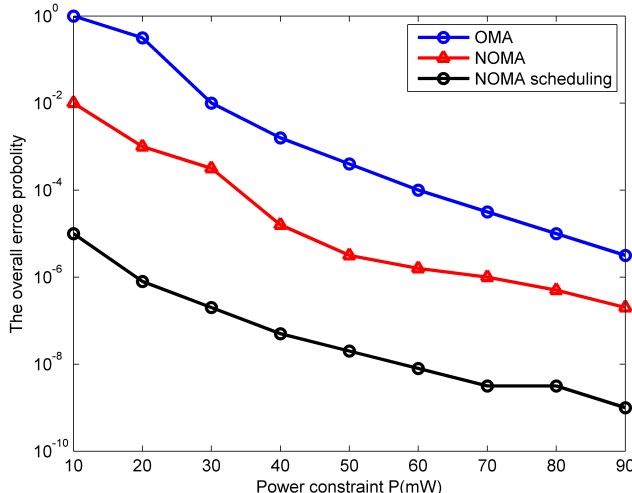

**Figure 5.** The overall error probability $\varepsilon_{BS,CU_1}$ versus the power, when M = 100 symbols.

In Figure 6, the effects of the total channel block length on $\varepsilon_{BS,CU_1}$ are studied. In this figure, we observe that $\varepsilon_{BS,CU_1}$ decreases monotonically with M. The implementation of NOMA and link scheduling in uRLLC can reduce the delay, which further demonstrates the superiority of the NOMA scheduling scheme. Similarly, the NOMA scheduling scheme delivered the best performance.

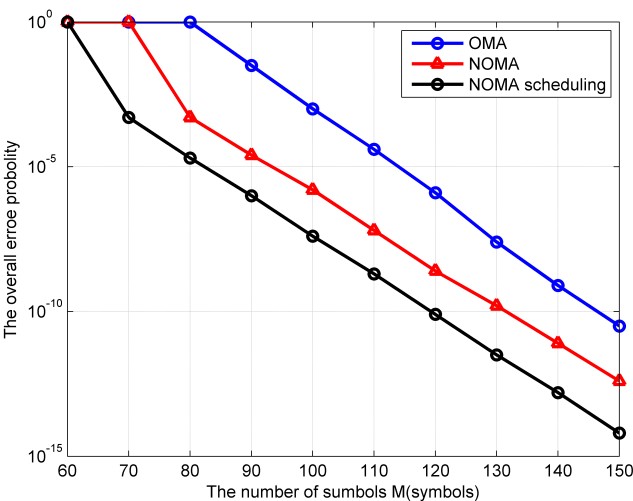

**Figure 6.** The overall error probability $\varepsilon_{\mathrm{BS,CU_1}}$ versus the channel blocklength, when P = 40 mW.

## 6. Conclusions

We examined the UAV scheduling and resource allocation issues of DL NOMA in ultra-reliable and low-latency UAV communication in this paper, and we implemented the stringent QoS specifications of uRLLC. According to the simulation results, the DLS scheduling algorithm performed significantly better than the GHW method, and the NOMA scheduling scheme performed better than both OMA and NOMA in terms of the overall error probability. In order to increase the performance of uRLLC and the spectrum efficiency, in the future, we will investigate link scheduling methods that are more suitable for NOMA in combination with UAVs.

**Author Contributions:** Conceptualization, X.L. and X.X.; methodology, K.Y.; software, X.X.; validation, X.L., X.X. and K.Y.; formal analysis, X.X.; investigation, X.X.; resources, X.L.; data curation, X.X.; writing—original draft preparation, X.L.; writing—review and editing, K.Y.; visualization, X.X.; supervision, X.X.; project administration, X.L.; funding acquisition, X.L. and K.Y. All authors have read and agreed to the published version of the manuscript.

**Funding:** This work was supported by the Natural Science Foundation of Shandong Province with Grants ZR2021QF050, ZR2021MF075 and ZR2022MF304; Shandong Natural Science Foundation Major Basic Research with Grant ZR2019ZD10; Shandong Key Research and Development Program with Grant 2019GGX1050; and Shandong Major Agricultural Application Technology Innovation Project with Grant SD2019NJ007.

**Institutional Review Board Statement:** Not applicable.

**Informed Consent Statement:** Not applicable.

**Data Availability Statement:** Not applicable.

**Conflicts of Interest:** The authors declare no conflict of interest.

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
