# Peer review of "NOMA and UAV Scheduling for Ultra-Reliable and Low-Latency Communications"

_drones, doi:10.3390/drones7010041_

Round 1

Reviewer 1 Report

Paper has presented new work but it has very weak justification in terms of results and analysis. Author should add computational complexity analysis of the method and should be compare its performance with any relevant work in order to justify its worth.

Its has many grammer mistakes so paper needed to be improved in terms of english writing. Require proof reading before being further considered, such as

line 51, Faced with these challenges, the

line 61, soo many commas in one sentence

line 92, NOMA has extraordinary expected in

line 99, why first letter is capital in , Maximum Link Scheduling

line 125, repeated representation in abbreviation of BS and CU

line 290, the GHW strategy does. Because of the

The proposed algorithm  claimed that, the steps are optimized to the given workloads. However, the time complexity and space complexity not determined in the manuscript.

The  finding and limitations of the proposed work not defined in the work.

The future wok is not defined in the manuscript

The simulation results and implementation are not align to the considered problem.

There must be new references and simulation alignment as the baselines in the manuscript.

Author Response

Dear reviewer,

Thank you for your suggestions. All your suggestions are very important, and they have guided me in my thesis writing and scientific work. We have revised the manuscript based on your comments and carefully proofread it to minimize typographical, grammatical, and bibliographical errors.

The following is a description of our revisions based on your comments.

  1. Comment: The time complexity and space complexity not determined in the manuscript.

Response: Thank you for your comments on the manuscript. In order to enhance the soundness of our results and analysis, we have analyzed the total complexity of the system. First, the complexity of Algorithm 1, Algorithm 2 and the resource allocation scheme were analyzed separately, and the total complexity of the system was obtained from them.

  1. Comment: It has many grammatical errors, so the paper needs improvement in English writing.

Response: We have re-proofread our manuscript and corrected the grammatical errors.

  1. Comment: The work does not define the findings and limitations of the proposed work and does not define the future work

Response: According to your suggestion, we have defined the findings and limitations of the work in the introduction and related work. In the conclusion section, we summarize the work and define the future work.

  1. Comment: The simulation results and implementation are not consistent with the considered problem and the manuscript must have a new reference and simulation alignment as a baseline.

Response: Our simulation is divided into two parts, the first part is a comparison between our proposed UAV scheduling algorithm and GHW algorithm, and the second part is a comparison of the whole system, through the simulation we found that the proposed system can show better performance.

Thanks again for your suggestion, and I hope to learn more from you.

                                                                                       Sincerely yours,

                                                                                               Kan Yu

Reviewer 2 Report

The authors proposed a NOMA and UAV Scheduling for Ultra-reliable and Low-Latency Communications. Here are the comments:

1. To gain a better understanding for the reader, the author should mention what modulation format used in this simulation, if not then state the reason why the author choose to do so.

2. In this manuscript the author mentioned that NOMA having a high spectral efficiency. What is the spectral efficiency for this system?

3. The SIC process are gain more noise onto demodulation process, please explain how about this system?

Author Response

Dear reviewer,

Thank you for your suggestions. All your suggestions are very important, and they have guided me in my thesis writing and scientific work. We have revised the manuscript based on your comments and carefully proofread it to minimize typographical, grammatical, and bibliographical errors.

The following is a description of our revisions based on your comments.

1.Comment: For better understanding by the reader, the authors should mention the modulation format used in this simulation and, if not then state the reason why the author choose to do so.

Response: Thank you for your comments on the manuscript. The modulation format that we use is FDD.

2.comment: In this manuscript, the authors mention that NOMA has high spectral efficiency. What is the spectral efficiency of this system?

Response: NOMA can be used to implement uRLLC by reducing the transmission error rate and lowering the transmission delay. In this paper, we mainly consider the implementation of uRLLC through NOMA, which can improve the spectral efficiency, but it is not studied much in this paper.

3.Comment: The SIC process are gain more noise onto demodulation process, please explain how about this system?

Response: The system proposed in this paper introduces multiple UAVs, each of which serves a separate set of users. To avoid users from being interfered by non-associated UAVs, we propose a UAV scheduling algorithm. To implement uRLLC, we optimize the power control and channel block length allocation of the system by NOMA. The final simulation results show that our proposed system exhibits good performance.

Thanks again for your suggestion, and I hope to learn more from you.

                                                                                 Sincerely yours,

                                                                                         Kan Yu

Round 2

Reviewer 1 Report

all comments are addressed

Reviewer 2 Report

The author answer all the question give by the reviewer quite well. I suggest to accept this paper for its current form.